# SELF-ATTENTIVE RATIONALIZATION FOR GRAPH CONTRASTIVE LEARNING

## ABSTRACT

Graph augmentation is the key component to reveal instance-discriminative features of a graph as its rationale in graph contrastive learning (GCL). And existing rationale-aware augmentation mechanisms in GCL frameworks roughly fall into two categories and suffer from inherent limitations: (1) non-heuristic methods with the guidance of domain knowledge to preserve salient features, which require expensive expertise and lack generality, or (2) heuristic augmentations with a co-trained auxiliary model to identify crucial substructures, which face not only the dilemma between system complexity and transformation diversity, but also the instability stemming from the co-training of two separated sub-models. Inspired by recent studies on transformers, we propose Self-attentive Rationale guided Graph Contrastive Learning (SR-GCL), which integrates rationale generator and encoder together, leverages the self-attention values in transformer module as a natural guidance to delineate semantically informative substructures from both node- and edge-wise perspectives, and contrasts on rationale-aware augmented pairs. On real-world biochemistry datasets, visualization results verify the effectiveness of self-attentive rationalization, and the performance on downstream tasks demonstrates the state-of-the-art performance of SR-GCL for graph model pre-training.

## 1 INTRODUCTION

Graph augmentation is a crucial enabler for graph contrastive learning (GCL) (You et al., 2020; Qiu et al., 2020; Zhu et al., 2020). It pre-trains the model to yield instance-discriminative representations by contrasting augmented samples against each other, without hand-annotated labels. To achieve this goal, early studies (You et al., 2020; 2021; Qiu et al., 2020; Zhu et al., 2020) conduct random corruptions in topological structures (*i.e.,* nodes and edges) or attributes to construct contrastive pairs. However, such random corruptions, especially on salient substructures, easily cause a semantic gap between two augmented views of the same anchor graph, misguiding the following contrastive optimization procedure (Wang et al., 2021; Li et al., 2022).

To mitigate this, there has been recent interest in rationale discovery (Chang et al., 2020; Suresh et al., 2021; Li et al., 2022) as graph augmentation. We systematize these studies as rationale-aware augmentations, where a rationale exhibits a graph's instance-discriminative information from the others. The dominant paradigm often consists of two subsequent modules: the rationale discovery function and the rationale encoder, which aim at creating the rationale-aware views and yielding their representations to contrast, respectively. To find rationales, early studies turn to domain knowledge to highlight the salient parts of graphs (Zhu et al., 2021; Liu et al., 2022). For instance, Rong et al. (2020) leverage RDkit (Landrum, 2010), an assistant software of chemistry, to capture crucial functional groups with high activity in molecule graphs. However, such expertise is expensive or even inaccessible in some scenarios (Tang et al., 2014). Besides, bringing in too much prior knowledge might harm generalization (Wang et al., 2022). To mitigate this problem, recent efforts (Suresh et al., 2021; Li et al., 2022) introduce an auxiliary model instead to automatically identify rationales, which is named the rationale generator and co-train with the rationale encoder. In this ad-hoc scheme, however, we reveal two inherent limitations:

- Typically, the generator is tailor-made for one single transformation of graph data (Suresh et al., 2021; Li et al., 2022), forcing the focus on either node- or edge-wise rationales (*e.g.,* Figures 1(b)

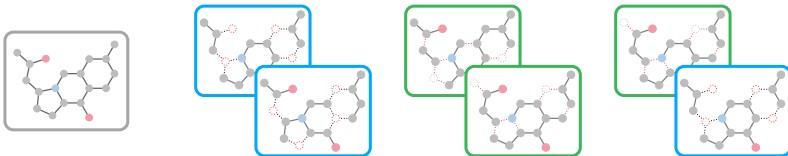

| (a) Original graph | (b) Node-wise views | (c) Edge-wise views | (d) Dual views |

Figure 1: Rationale-aware graph augmentation preserves instance-discriminative features in original graph (*e.g.,* (a)). Existing frameworks tailor-make an auxiliary model for one single transformation (*e.g.,* (b) or (c)). SR-GCL constructs both node- and edge-wise rationale-aware views (*e.g.,* (d)).

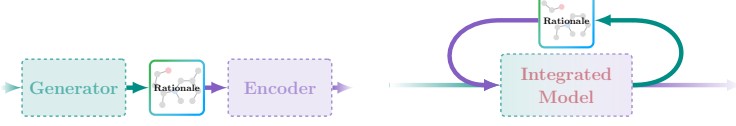

| (a) Separate generator-encoder scheme | (b) Integrated scheme |

Figure 2: Separate generator-encoder in (a) may lead to instability and biased representation, while SR-GCL integrates them together in (b) to pursue high-quality rationales and representations.

and 1(c)). The lack of view diversity confines the rationale-aware augmentations to one transformation, while leaving the cross-transformation untouched. Worse still, it might degenerate the effectiveness of contrastive learning, as the studies (Chen et al., 2020; You et al., 2020) empirically show that "no single transformation suffices to learn good representations". Here we ascribe this crux of generator to the lack of transformation diversity, and argue that a high-performing generator is supposed to be equipped with perspectives of both node and edge (*e.g.,* Figure 1(d)).

• As illustrated in Figure 2(a), the generator, aiming at discovering rationales, separates from the subsequent encoder, which specializes in encoding them. While conceptually appealing, we hypothesize that these separate modules cooperate with each other to pursue high-quality rationales unsmoothly. Because the supervision signal for the generator is remotely generated by the contrastive optimization of the encoder, much of which is weak. Moreover, co-optimizing two sub-models could make the pre-training more complicated and time-consuming but less stable.

To resolve these limitations, we draw inspiration from the transformers (Vaswani et al., 2017) to reshape the generator-encoder scheme. Despite originally being proposed for language (Devlin et al., 2019) and vision tasks (Dosovitskiy et al., 2021), transformers are attracting a surge of interest in graph area (Wu et al., 2021; Chen et al., 2022; Rampásek et al., 2022). At the core is the self-attention operation, which models pairwise connections between tokens and yields high-quality representations. We find self-attention de facto a natural mechanism to concurrently discover and condense rationale information from both edge- and node-wise transformations. By prepending a special token as its proxy and treating its nodes as other tokens, self-attention is able to elegantly indicate the importance of each node and each edge (See Section 2.2 and Figure 4). Sampling nodes and edges based on the importance scores (*i.e.,* heterogeneous transformations) allows us to generate both the node- and edge-wise subgraphs (*i.e.,* rationales) simultaneously. Moreover, self-attention can directly output the rationale representations without additional modules. In stark contrast to the prior generator-encoder scheme, the "self-attentive rationalization" not only accomplishes diverse rationales in one shot, but also integrates the functions of rationale discovery and encoding together.

With the Self-attentive Rationalization as graph augmentation, we incorporate it into GCL and name the framework **SR-GCL**. Specifically, two augmented views stem from the node- and edge-wise rationales, respectively; subsequently, the contrastive optimization pulls close representations of contrastive pair augmented from the same anchor graph and pushes away those of different anchors by minimizing contrastive loss. Compared with conventional GCL methods, our SR-GCL collates and conflates the instance-discriminative information across both node- and edge-wise transformations to construct rationale-aware contrastive pairs from dual perpectives. Henceforth, we find that such strategy improves the generalization performance of pre-trained model on downstream tasks, while simultaneously interpreting the contribution of each node/edge to instance-discrimination. Extensive experiments show that SR-GCL sets the new state-of-the-art for graph pre-training across a number of biochemical molecule and social network benchmark datasets in (Wu et al., 2018a; Morris et al., 2020). Codes are available at https://anonymous.4open.science/r/SR-GCL-EDD3.

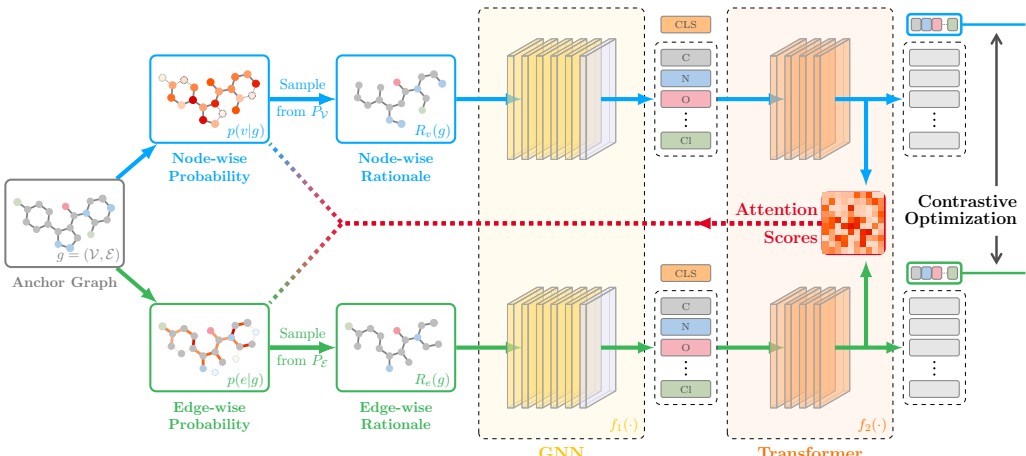

Figure 3: SR-GCL framework, which constructs node- and edge-wise rationales (*i.e.,* $R_v(g)/R_e(g)$) by sampling from corresponding probability distributions (*i.e.,* $P_\mathcal{V}(\cdot|G)/P_\mathcal{E}(\cdot|G)$). Then contrastive optimization is performed to encourage the agreement between views of the same anchor.

## 2 METHODOLOGY

Scrutinizing leading GCL methods (You et al., 2020; 2021; Qiu et al., 2020; Zhu et al., 2020; Suresh et al., 2021; Li et al., 2022), we can summarize the dominant scheme as a combination of two subsequent modules: graph augmentation and contrastive optimization. During the pre-training phase, the cooperation of these modules enables the backbone encoder to learn how to discover and parameterize the instance-discriminative information for graph instances. Hereafter, the pre-trained encoder is fine-tuned on downstream tasks. See Appendix A.1 for the literature review of GCL.

Preserving instance-discriminative semantics when augmenting graphs has been founded critical in recent studies (Suresh et al., 2021; Li et al., 2022). Here we systemize an augmented view holding such salient semantics as a rationale (Chang et al., 2020; Li et al., 2022), which can help discriminate a graph instance from the other instances. However, these studies focus mainly on discovering homogeneous rationales from a single perspective of either node or edge, thereby easily resulting in sub-optimal performance (Chen et al., 2020; You et al., 2020). To explore more effective and diverse rationales, we propose a novel transformer-based pre-training framework, SR-GCL. As Figure 3 shows, it uses the graph transformer as the backbone encoder, whose inherent self-attention map allows us to discover heterogeneous rationales from dual perspectives of both node and edge; subsequently, upon the node- and edge-wise rationales, it conducts the contrastive optimization to pre-train the backbone. Appendix B presents the detailed algorithm of SR-GCL.

### 2.1 BACKBONE MODEL: GRAPH TRANSFORMER

We begin by describing our backbone model, which is the cornerstone of SR-GCL. Distinguishing from the current GCL studies that employ "vanilla" graph neural networks (GNNs) (Thomas et al., 2022) as the backbone model being pre-trained and fine-tuned, we use the graph transformer instead (Wu et al., 2021; Rampásek et al., 2022; Chen et al., 2022). Specifically, it adopts the transformer architecture (Vaswani et al., 2017; Dosovitskiy et al., 2021) to process the structural input of graph (See Appendix A.2 for the literature review). It consists of (1) a GNN component, which creates node representations of a graph by propagating information between the locally adjacent nodes and then converts them into tokens; and (2) a transformer component, which refines the token representations by applying the self-attention mechanism to model the global connections between any two nodes. Next, we will elaborate on the graph transformer backbone.

**GNN Component:** Formally, let $g = (\mathcal{V}, \mathcal{E})$ be a graph instance, where $\mathcal{V}$ and $\mathcal{E}$ are its node and edge sets, respectively. Taking $g$ as input, this component first leverages a vanilla GNN $f_1$ to yield the node representations $\mathbf{X} \in \mathbb{R}^{|\mathcal{V}| \times d}$, and then converts them into a series of tokens $\mathbf{Z} \in \mathbb{R}^{(|\mathcal{V}|+1) \times d}$:

$$\mathbf{X} = [\mathbf{x}_1, \mathbf{x}_2, \cdots, \mathbf{x}_{|\mathcal{V}|}] = f_1(g), \quad \mathbf{Z} = [\mathbf{x}_1, \mathbf{x}_2, \cdots, \mathbf{x}_{|\mathcal{V}|}, \mathbf{x}_{\text{CLS}}], \tag{1}$$

where $\mathbf{x}_v \in \mathbb{R}^d$ is the $d$-dimensional representation of node $v \in \mathcal{V}$, and all node representations are viewed as a sequence. It is worth highlighting that we append a special trainable $\langle\text{CLS}\rangle$ token to this sequence, which serves as the graph-level representation for contrastive optimization in pre-training and classification in downstream fine-tuning. That is, $\langle\text{CLS}\rangle$ can be viewed as a virtual supernode connected with all nodes, and expands $\mathcal{V}$ to $\mathcal{V}^* = \mathcal{V} \cup \{\langle\text{CLS}\rangle\}$.

**Transformer Component:** On the tokens, we build a transformer $f_2$ composed of $L$ layers:

$$\mathbf{Z}^L = [\mathbf{z}_1^L, \mathbf{z}_2^L, \cdots, \mathbf{z}_{|\mathcal{V}|}^L, \mathbf{z}_{\text{CLS}}^L] = f_2(\mathbf{Z}), \tag{2}$$

where $\mathbf{Z}^L \in \mathbb{R}^{(|\mathcal{V}|+1) \times d}$ collects the $d$-dimensional representations of all tokens. Now, we dive into the core of $f_2$ — the self-attention mechanism that establishes the pairwise interactions across all tokens so as to enrich representations with the global information. Formally, for the pair of tokens $v$ and $u$, the attention score $a_{v,u}^l$ in the $l$-th transformer layer can be obtained as follows:

$$a_{v,u}^l = \frac{\kappa(\mathbf{z}_v^{l-1}, \mathbf{z}_u^{l-1})}{\sum_{w \in \mathcal{V}^*} \kappa(\mathbf{z}_v^{l-1}, \mathbf{z}_w^{l-1})}, \quad \kappa(\mathbf{z}_v^{l-1}, \mathbf{z}_u^{l-1}) := \exp\frac{\langle \mathbf{W}_Q^l \mathbf{z}_v^{l-1}, \mathbf{W}_K^l \mathbf{z}_u^{l-1} \rangle}{\sqrt{d}}, \tag{3}$$

where $\mathbf{z}_v^{l-1}$ is the representation of token $v$ after $(l-1)$ layers, and $\mathbf{z}_v^0$ is set as $\mathbf{z}_v$ in $\mathbf{Z}$ at the initial step; $\kappa(\cdot, \cdot)$ is the exponential kernel function with the dot product operation $\langle\cdot, \cdot\rangle$; $\mathbf{W}_Q^l / \mathbf{W}_K^l \in \mathbb{R}^{d \times d}$ are the trainable query/key projection matrices. Attending to all possible tokens, the representation of token $v$ at the $l$-th layer can be updated as follows:

$$\mathbf{z}_v^l = \sum_{u \in \mathcal{V}^*} a_{v,u}^l \mathbf{W}_V^l \mathbf{z}_u^{l-1}, \tag{4}$$

where $a_{v,u}^l$ indicates the contribution of token $u \in \mathcal{V}^*$ to token $v$; $\mathbf{W}_V^l \in \mathbb{R}^{d \times d}$ is the trainable value projection matrix. After $L$ layers, the representations are recursively updated to $\mathbf{Z}^L$ in Equation 2.

**Why Graph Transformer?** Most vanilla GNN-based backbones are responsible solely to the representation learning, while leaving the graph augmentation step to an additional non-parametric (*e.g.,* corrupt graphs randomly or based on domain knowledge) or parametric (*e.g.,* generate rationales via an attention or masking network) module. Here we focus on the line of parametric rationale-aware augmentation. Clearly, this augmentation module is disjoint from these conventional backbones. Concretely, graph transformer backbone allows us to integrate the representation learning and rationale generation together. Next, we will introduce a simple yet effective rationalization strategy based on the self-attention mechanism of the transformer.

## 2.2 GRAPH AUGMENTATION: SELF-ATTENTIVE RATIONALIZATION

We now present our augmentation strategy — self-attentive rationalization, which exploits the self-attention mechanism to generate rationales from two perspectives of node and edge in one shot.

### 2.2.1 SELF-ATTENTION SCORES FOR RATIONALIZATION

For a graph instance, we reshape the self-attention values in the first transformer layer into a 2-D heat map as illustrated in Figure 4, which enables us to estimate the instance-discriminative power of each node and edge — that is, the influence of a certain node or edge on discriminating the graph instance from all the others. Specifically, for node $v \in \mathcal{V}$, its influence score $p_v$ is derived from the normalized attention score between it and the $\langle\text{CLS}\rangle$ token; meanwhile, for edge $e = (v, u) \in \mathcal{E}$, the pairwise interaction between endpoints $v$ and $u$ naturally depicts its importance $p_e$:

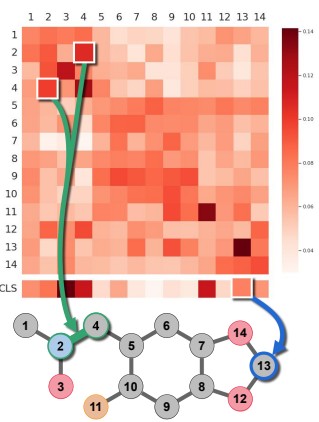

$$p_v = \frac{\kappa(\mathbf{z}_{\text{CLS}}, \mathbf{z}_v)}{\sum_{v' \in \mathcal{V}} \kappa(\mathbf{z}_{\text{CLS}}, \mathbf{z}_{v'})}, \quad p_e = \frac{\kappa(\mathbf{z}_v, \mathbf{z}_u)}{\sum_{e' \in \mathcal{E}} \kappa(\mathbf{z}_{v'}, \mathbf{z}_{u'})}, \tag{5}$$

where $\kappa(\cdot, \cdot)$ is the exponential kernel function defined in Equation 3; $\kappa(\mathbf{z}_{\text{CLS}}, \mathbf{z}_v)$ evaluates node $v$'s attribution to the

Figure 4: Self-attention map.

graph-level representation, *i.e.,* the $\langle$CLS$\rangle$ token; $\kappa(\mathbf{z}_v, \mathbf{z}_u)$ reflects the volume of information propagated on edge $e = (v, u)$, which is further normalized over all edges $e' = (v', u') \in \mathcal{E}$. With the contributions of each node and edge to instance-discrimination in hand, we can further conduct probabilistic sampling to generate both node- and edge-wise rationales for contrastive optimization.

### 2.2.2 NODE- & EDGE-WISE RATIONALE GENERATION

We first approximate the distribution of rationale-aware views, and then conduct probabilistic sampling to maintain the diversity of views. With a slight abuse of notation, uppercase $G$ and $R(G)$ separately represent a random variable of graph and its rationale, while lowercase $g$ and $R(g)$ are their samples correspondingly. Conditional probabilistic function and conditional probabilistic distribution are denoted by $p(\cdot|\cdot)$ and $P(\cdot|\cdot)$, respectively.

**Distribution of Node- & Edge-wise Rationales.** Given a graph $g$, the probability distribution of its node-wise rationale-aware view $P_1$ is approximated as:

$$P_1(R(G) = R(g)|G = g) = \prod_{v \in \mathcal{V}_R} p(v|g) \prod_{v \in \mathcal{V} \backslash \mathcal{V}_R} (1 - p(v|g)), \quad (6)$$

where $\mathcal{V}$ and $\mathcal{V}_R$ are the node sets of $g$ and rationale $R(g)$, respectively; $p(v|g)$ denotes the probability of $v$ being included into $R(g)$, which quantizes how semantically informative it is. Similarly, the edge-wise probability distribution $P_2$ can be formulated as:

$$P_2(R(G) = R(g)|G = g) = \prod_{e \in \mathcal{E}_R} p(e|g) \prod_{e \in \mathcal{E} \backslash \mathcal{E}_R} (1 - p(e|g)), \quad (7)$$

where $\mathcal{E}$ and $\mathcal{E}_R$ are the edge sets of $g$ and $R(g)$, respectively; $p(e|g)$ reflects the discriminative power of edge $e$. And $p(v|g)$ (or $p(e|g)$) is parameterized by $p_v$ (or $p_e$) in Equation 5.

**Sampling with Fixed Ratios.** To construct anchor graph $g$'s node-wise rationale-aware view, we sample from $P_1(\cdot|G = g)$ under the constraint of a fixed node sampling ratio $\rho_v$:

$$R_1(g) \sim P_1(\cdot|G = g) \quad \text{s.t.} \ |\mathcal{V}_R| = \rho_v \cdot |\mathcal{V}|. \quad (8)$$

After obtaining sampled nodes, the edges between them are kept to construct a complete subgraph. Analogically, edge-wise rationale-aware view can be generated with another edge sampling ratio $\rho_e$:

$$R_2(g) \sim P_2(\cdot|G = g) \quad \text{s.t.} \ |\mathcal{E}_R| = \rho_e \cdot |\mathcal{E}|, \quad (9)$$

where the endpoints of sampled edges are preserved, while the others are discarded.

Without involving any auxiliary models, two rationale-aware views with different transformations (*i.e.,* node- and edge-wise) are generated under the guidance of self-attention rationalization.

### 2.3 RATIONALE REPRESENTATION LEARNING AND CONTRASTIVE OPTIMIZATION

With the rationale-aware views (*i.e.,* $R_1(g)/R_2(g)$) generated by the the self-attentive rationalization process in hand, now we follow the pipeline of representation learning and contrastive optimization to empower the backbone graph transformer model with instance-discriminative ability.

### 2.3.1 RATIONALE REPRESENTATION LEARNING

As illustrated in Figure 3, the GNN $f_1$ and transformer $f_2$ sequentially process the rationale-aware views as in Equation 1 and Equation 2 to yield their representations:

$$\mathbf{Z}_1^L = f_2(f_1(R_1(g))), \qquad \mathbf{Z}_2^L = f_2(f_1(R_2(g))). \quad (10)$$

Subsequently, the $\langle$CLS$\rangle$ tokens in $\mathbf{Z}_1^L$ and $\mathbf{Z}_2^L$, which we denote as $\mathbf{z}_{1,\text{CLS}}^L$ and $\mathbf{z}_{2,\text{CLS}}^L$ separately, are picked out as the graph-level representations of node- and edge-wise views (*i.e.,* $R_1(g)/R_2(g)$). Then we follow the commonly-adopted projection operation to project them into a latent space through a MLP-based projection head $h$ with $l_2$ norm:

$$\mathbf{r}_1 = h(\mathbf{z}_{1,\text{CLS}}^L), \qquad \mathbf{r}_2 = h(\mathbf{z}_{2,\text{CLS}}^L). \quad (11)$$

The transformer module plays the roles of both the rationale generator, whose attention map guides rationale-aware augmentations of both node- and edge-wise views, and the encoder, who generates graph representations for subsequent contrastive optimization.

Figure 5: (a): A molecule graph. (b): Random attention. (c): Local attention of adjacency matrix. (d): Global attention. (e): GraphBigBird, where white entry indicates absence of attention.

### 2.3.2 CONTRASTIVE OPTIMIZATION

The Info-NCE loss (van den Oord et al., 2018) is adopted to maximize the mutual information between node- and edge-wise rationale-aware views of the same anchor graph:

$$\min_{f_1, f_2, h} \mathcal{L} = \mathbb{E}_{g \in \mathcal{G}} \, l(g), \qquad l(g) = -\log \frac{\exp\left(\mathbf{r}_1^\top \mathbf{r}_2 / \tau\right)}{\exp\left(\mathbf{r}_1^\top \mathbf{r}_2 / \tau\right) + \sum_{\mathbf{r}' \in \mathcal{R}'} \exp\left(\mathbf{r}_1^\top \mathbf{r}' / \tau\right)}, \qquad (12)$$

where $\tau$ denotes the temperature parameter, $\mathbf{r}_1$ and $\mathbf{r}_2$ are the rationale-aware views of the same anchor graph (*i.e.,* positive views derived from Equation 11), and $\mathcal{R}'$ summarizes the representations of the other graphs' views in the minibatch (*i.e.,* negative views).

The supervision signal guides the backbone model (*i.e.,* both GNN and transformer module) to learn instance-discriminative representations, which indicates that the self-attention mechanism in transformer is enhanced concurrently, thus bring about more accurate self-attentive rationalization.

Finally, after pre-training, the projection head $h$ will be thrown away. And it's worth mentioning that SR-GCL can be easily adapted to any transformer-based graph model with self-attention mechanism.

### 2.4 DISCUSSION ON TRANSFORMER COMPLEXITY: GRAPHBIGBIRD

Compared with conventional GNN model with a time and space complexity of $O(N)$ *w.r.t.* the number of nodes, the graph transformer with self-attention mechanism is empowered with powerful representation ability but a quadratic complexity *w.r.t.* the length of token sequence (*i.e.,* the number of nodes) as well. As evaluated in (Wu et al., 2021; Rampásek et al., 2022), such quadratic complexity just adds acceptable computation overhead when processing graphs with only dozens of nodes. Nevertheless, in social network datasets, graph instances could hold thousands of nodes. Thus, the graph transformer, especially the self-attention mechanism would not scale well to such large-scale graphs, due to the quadratic complexity of pairwise attention. To mitigate this, we get inspiration from the sparse attention mechanism of BigBird (Zaheer et al., 2020) in the NLP area and propose GraphBigBird tailor-made for the graph transformer.

**GraphBigBird.** As illustrated in Figure 5, GraphBigBird contains three parts: (1) random attention with a hyperpatameter $r$ controlling the sparsity, (2) local attention corresponding to the adjacency matrix of the graph sample, and (3) global attention connecting each token with the ⟨CLS⟩ token. The major difference between BigBird and GraphBigBird lies on the local attention part. Concretely, for a sequence of NLP tokens, BigBird adopts a sliding window to capture the proximity between a token and its neighboring tokens. GraphBigBird adapts this idea to graph area by replacing the sliding window with the adjacency matrix, as Figure 5(c) shows. This simple strategy is effective to depict the local connectivity of graph.

With GraphBigBird, SR-GCL, requiring the graph transformer to be the backbone, is able to break the graph scale limitation of dozens of nodes and generalize to domains containing large-scale graph data, to demonstrate its universal effectiveness.

## 3 EXPERIMENTS

In this section, extensive experiments are conducted on biochemical molecule and social network datasets with transfer and unsupervised learning settings to demonstrate the state-of-the-art per-

Table 1: Transfer learning. Test ROC-AUC scores (%) of pre-trained models on downstream datasets. G represents the commonly used GIN model in previous baselines, while T is the graph transformer model containing both GNN and transformer module depicted in this paper. Statistics of G models are from their original papers and those of T models are obtained by replanting the graph transformer model into their official released codes. red and blue indicate the best and the second best performance on each dataset, respectively.

| Dataset | | BBBP | Tox21 | ToxCast | SIDER | ClinTox | MUV | HIV | BACE | AVG. | GAIN |
|---|---|---|---|---|---|---|---|---|---|---|---|
| No Pre-Train | G | 65.8 ±4.5 | 74.0 ±0.8 | 63.4 ±0.6 | 57.3 ±1.6 | 58.0 ±4.4 | 71.8 ±2.5 | 75.3 ±1.9 | 70.1 ±5.4 | 67.0 | - |
| | T | 68.12±2.94 | 73.48±0.32 | 62.67±0.56 | 59.44±1.26 | 69.45±3.12 | 72.18±1.16 | 74.41±1.45 | 74.23±4.62 | 68.24 | 1.24 |
| Infomax | G | 68.8 ±0.8 | 75.3 ±0.5 | 62.7 ±0.4 | 58.4 ±0.8 | 69.9 ±3.0 | 75.3 ±2.5 | 76.0 ±0.7 | 75.9 ±1.6 | 70.3 | 3.3 |
| | T | 69.15±0.68 | 74.62±0.45 | 61.85±0.75 | 60.14±0.72 | 74.78±2.66 | 74.83±1.52 | 74.95±1.26 | 76.32±1.78 | 70.83 | 3.83 |
| EdgePred | G | 67.3 ±2.4 | 76.0 ±0.6 | 64.1 ±0.6 | 60.4 ±0.7 | 64.1 ±3.7 | 74.1 ±2.1 | 76.3 ±1.0 | 79.6 ±1.2 | 70.3 | 3.3 |
| | T | 69.32±1.24 | 76.78±0.49 | 64.43±0.72 | 60.15±0.62 | 73.61±2.49 | 72.41±1.26 | 75.87±0.82 | 78.46±1.87 | 71.37 | 4.37 |
| AttrMasking | G | 64.3 ±2.8 | 76.7 ±0.4 | 64.2 ±0.5 | 61.0 ±0.7 | 71.8 ±4.1 | 74.7 ±1.4 | 77.2 ±1.1 | 79.3 ±1.6 | 71.1 | 4.1 |
| | T | 66.42±3.16 | 77.22±0.67 | 64.89±0.89 | 60.88±0.45 | 77.28±3.88 | 74.12±1.98 | 77.23±1.51 | 79.56±2.13 | 72.20 | 5.20 |
| ContextPred | G | 68.0 ±2.0 | 75.7 ±0.7 | 63.9 ±0.6 | 60.9 ±0.6 | 65.9 ±3.8 | 75.8 ±1.7 | 77.3 ±1.0 | 79.6 ±1.2 | 70.9 | 3.9 |
| | T | 68.52±1.12 | 75.29±0.56 | 63.23±0.46 | 61.29±0.75 | 74.90±3.18 | 70.92±2.36 | 76.82±1.13 | 80.16±1.84 | 71.39 | 4.39 |
| GraphCL | G | 69.68±0.67 | 73.87±0.66 | 62.40±0.57 | 60.53±0.88 | 75.99±2.65 | 69.80±2.66 | 78.47±1.22 | 75.38±1.44 | 70.77 | 3.77 |
| | T | 70.41±1.06 | 73.82±1.04 | 63.06±0.41 | 60.38±1.28 | 77.79±3.02 | 73.81±2.02 | 75.62±0.89 | 78.28±1.08 | 71.64 | 4.64 |
| GraphLoG | G | 72.5 ±0.8 | 75.7 ±0.5 | 63.5 ±0.7 | 61.2 ±1.1 | 76.7 ±3.3 | 76.0 ±1.1 | 77.8 ±0.8 | 83.5 ±1.2 | 73.4 | 6.4 |
| | T | 67.59±1.58 | 75.95±0.80 | 63.63±0.57 | 59.85±2.11 | 79.10±3.24 | 72.84±1.79 | 72.54±1.60 | 83.55±1.96 | 71.88 | 4.88 |
| AD-GCL | G | 70.01±1.07 | 76.54±0.82 | 63.07±0.72 | 63.28±0.79 | 79.78±3.52 | 72.30±1.61 | 78.28±0.97 | 78.51±0.80 | 72.72 | 5.72 |
| | T | 67.88±0.97 | 72.97±1.19 | 63.19±0.44 | 60.18±0.95 | 78.83±1.60 | 75.62±2.54 | 75.54±1.29 | 73.37±1.55 | 70.95 | 3.95 |
| JOAO | G | 70.22±0.98 | 74.98±0.29 | 62.94±0.48 | 59.97±0.79 | 81.32±2.49 | 71.66±1.43 | 76.73±1.23 | 77.34±0.48 | 71.90 | 4.90 |
| | T | 70.78±0.62 | 75.25±1.46 | 63.79±0.84 | 61.04±0.87 | 78.68±2.97 | 75.67±2.57 | 76.98±1.16 | 77.49±1.23 | 72.46 | 5.46 |
| RGCL | G | 71.42±0.66 | 75.20±0.34 | 63.33±0.17 | 61.38±0.61 | 83.38±0.91 | 76.66±0.99 | 77.90±0.80 | 76.03±0.77 | 73.16 | 6.16 |
| | T | 71.36±0.85 | 75.72±0.40 | 63.93±0.36 | 60.91±0.56 | 80.02±1.59 | 75.94±1.19 | 77.81±0.62 | 79.92±1.02 | 73.20 | 6.20 |
| SR-GCL edge[1] | T | 71.62±1.87 | 74.98±0.96 | 64.87±0.65 | 61.32±1.59 | 80.29±2.68 | 75.11±1.63 | 77.64±1.68 | 79.69±0.87 | 73.19 | 6.19 |
| SR-GCL node[1] | T | 73.01±0.94 | 75.12±1.68 | 65.69±1.02 | 61.21±0.93 | 80.47±3.55 | 77.92±3.12 | 77.89±0.68 | 81.33±0.92 | 74.08 | 7.08 |
| SR-GCL dual[1] | T | 72.60±0.89 | 77.30±0.84 | 65.86±0.61 | 61.46±1.36 | 81.84±1.60 | 77.76±1.18 | 78.52±0.74 | 81.58±0.97 | 74.62 | 7.62 |

[1] SR-GCL with edge-wise perspective, node-wise perspective and dual perspectives of both node and edge.

formance of SR-GCL. Also, we visualize the rationales to illustrate the effectiveness of the self-attentive rationalization mechanism. Finally, due to the limitation of space, the analysis for hyper-parameter sensitivity is provided in Appendix G.

## 3.1 TRANSFER LEARNING

**Setup and baselines.** Following the commonly adopted transfer learning settings and evaluation metrics in (Hu et al., 2020), the backbone model is pre-trained on large scale label-free dataset – ZINC-2M (Sterling & Irwin, 2015) and fine-tuned on downstream datasets with graph-level classification tasks in MoleculeNet (Wu et al., 2018a) to evaluate the transferability of schemes. SR-GCL is compared with competitive graph pre-training baselines, including Infomax (Velickovic et al., 2019) , EdgePred (Hamilton et al., 2017), AttrMasking Hu et al. (2020), ContextPred (Hu et al., 2020), GraphCL (You et al., 2020), GraphLoG (Xu et al., 2021), AD-GCL (Suresh et al., 2021), JOAO (You et al., 2021) and RGCL (Li et al., 2022) The statistics of datasets and briefs of baselines are presented in Appendix C and D, respectively.

**Performance on downstream tasks.** An MLP-based classifier is appended after the pre-trained backbone model and they are subsequently fine-tuned together on downstream graph classification tasks. In Table 1, we present all statistics from their original papers with the backbone model of GIN (Xu et al., 2019), which is denoted as G. To make a fair comparison, we also re-run all baselines with the same graph transformer model depicted in this paper, which we denote as T. With one pre-trained model, on each downstream dataset, we run fine-tuning procedures 10 times with random seeds from 0 to 9 to collect the test ROC-AUC of the epoch with the highest validation score, the mean and standard deviation statistics of which are showcased. Noted that the sparse attention mechanism – GraphBigBird is not adopted in this transfer learning experiment since molecule graphs only contain dozens of nodes. The detailed model structures and the computational efficiency comparison of GIN

Table 2: Unsupervised learning. Test accuracies (%) on multiple biochemical and social network datasets. Statistics are from their original papers except GraphMAE. red and blue indicate the best and the second best performance on each dataset, respectively.

| Dataset | NCI1 | PROTEINS | DD[1] | MUTAG | COLLAB | RDT-B[1] | RDT-M5K[1] | IMDB-B | AVG. | GAIN |
|---|---|---|---|---|---|---|---|---|---|---|
| No Pre-Train | 65.40±0.17 | 72.73±0.51 | 75.67±0.29 | 87.39±1.09 | 65.29±0.16 | 76.86±0.25 | 48.48±0.28 | 69.37±0.37 | 70.15 | - |
| graph2vec | 73.22±1.81 | 73.30±2.05 | - | 83.15±9.25 | - | 75.28±1.03 | 47.86±0.26 | 71.10±0.54 | - | - |
| InfoGraph | 76.20±1.06 | 74.44±0.31 | 72.85±1.78 | 89.01±1.13 | 70.05±1.13 | 82.50±1.42 | 53.46±1.03 | 73.03±0.87 | 74.02 | 3.87 |
| GraphCL | 77.87±0.41 | 74.39±0.45 | 78.62±0.40 | 86.80±1.34 | 71.36±1.15 | 89.53±0.74 | 55.99±0.28 | 71.14±0.44 | 75.71 | 5.56 |
| JOAO | 78.36±0.53 | 74.07±1.10 | 77.40±1.15 | 87.67±0.79 | 69.33±0.34 | 86.42±1.45 | 56.03±0.27 | 70.83±0.25 | 75.01 | 4.86 |
| AD-GCL | 75.86±0.62 | 75.04±0.48 | 75.73±0.51 | 88.62±1.27 | 74.89±0.90 | 92.35±0.42 | 56.24±0.39 | 71.49±0.98 | 76.28 | 6.13 |
| RGCL | 78.14±1.08 | 75.03±0.43 | 78.86±0.48 | 87.66±1.01 | 70.92±0.65 | 90.34±0.58 | 56.38±0.40 | 71.85±0.84 | 76.15 | 6.00 |
| GraphMAE[2] | 76.12±1.45 | 74.69±0.74 | 77.15±1.17 | 87.21±0.88 | 73.16±2.48 | 87.76±1.08 | 54.98±1.27 | 71.61±0.52 | 75.33 | 5.18 |
| SR-GCL | 77.76±0.21 | 75.38±0.46 | 78.70±0.79 | 87.95±1.88 | 76.62±0.50 | 91.12±0.44 | 56.71±0.62 | 72.06±0.74 | 77.03 | 7.03 |

[1] GraphBigBird is adopted.
[2] The evaluation metrics and data pre-processing are different from the other baselines, thus it is reproduced with the same settings as others.

and graph transformer are included in Appendix E and F. The overall results are summarized in Table 1, and the following observations can be obtained:

1. **SR-GCL outperforms existing pre-training baselines.** On 8 downstream datasets, SR-GCL empowers the backbone graph transformer model with an average ROC-AUC score of 74.62% and performance gain of 6.38% over the same graph transformer model without pre-training. And compared with other competitive pre-training schemes, it achieves the best performance on 5 out of 8 datasets and 3 second bests – the two leading positions on BACE both belong to GraphLoG, thus SR-GCL can be seen as the second best pre-training scheme on this dataset. Such clear performance margin between SR-GCL and previous baselines strongly validates its effectiveness as a framework of self-supervised learning benefiting backbone's transferability.

2. **Benefits of rationale-aware augmentations.** Among multiple variants of GCL framework, GraphCL can be viewed as the vanilla one with random augmentations. With the shared motivation of preserving instance-discriminative substructures – rationales in anchor graphs, AD-GCL, RGCL and SR-GCL are proposed, but with different rationale discovery mechanisms. The consistent performance gain of them over GraphCL empirically demonstrates the crucial role of augmentation mechanism in revealing the rationales in anchor graphs.

3. **Effectiveness of the combined design of rationale generator and encoder.** RGCL leverages an external auxiliary model as the rationale generator to reveal semantically important nodes for graph augmentation. With the same graph transformer (T) as the backbone, the difference between RGCL(T) and SR-GCL with only node-wise perspective lies on the rationale generator. Compared with the detached design in RGCL, the self-attentive rationalization mechanism enables SR-GCL to combine the rationale generator and encoder. As shown in Table 1, the performance gain of SR-GCL with only node-wise perspective, which is 74.08%, over RGCL(T), which is 73.20%, verifies our claim that such combination benefits the model.

4. **Advantage of diverse perspectives.** Despite of the rationale-aware augmentations, SR-GCL with only node-/edge-wise perspective has limited diversity of augmented views. The complete SR-GCL with dual perspectives outperforms them, thus verifying the advantage, which is also consistent with the observations in (Chen et al., 2020; You et al., 2020).

**Summary.** The experiments of transfer learning demonstrate the promising performance of SR-GCL as a self-supervised GCL scheme. Concretely, the comparisons among GCL variants (*e.g.,* GraphCL, AD-GCL and RGCL) and ablated structures of SR-GCL (*e.g.,* SR-GCL with node-/edge-wise perspective and dual perspectives) verify the effectiveness of our major contributions: (1) combining the rationale generator and encoder and (2) collating and conflating the instance-discriminative information across node- and edge-wise transformations.

## 3.2 UNSUPERVISED LEARNING

We further evaluate SR-GCL in the unsupervised learning settings (Sun et al., 2020), which cover both biochemical and social network datasets from TU datasets (Morris et al., 2020). Several leading

Figure 6: Self-attentive rationalization. (a): A molecule graph. (b): Corresponding attention map. (c): Node- (top) and edge-wise (bottom) attention, where darker color indicates higher score. (d): The contribution of each atom to Gasteiger partial charge (top) and partition coefficient (bottom).

pre-traing schemes are selected as the baselines, including graph2vec (Narayanan et al., 2017), InfoGraph (Sun et al., 2020), GraphCL (You et al., 2020), JOAO (You et al., 2021), AD-GCL (Suresh et al., 2021), RGCL (Li et al., 2022) and GraphMAE (Hou et al., 2022). The dataset statistics and baseline briefs are available at Appendix C and D, respectively.

**GraphBigBird on large-scale graphs.** Before presenting the performance comparison, we would like to highlight the sparse attention mechanism — GraphBigBird — to reduce the computational complexity when applying the graph transformer for large-scale graphs. GraphBigBird, as depicted in Section 2.4, is adopted on three datasets containing graphs with thousands of nodes: DD, RDT-B and RDT-M. The detailed model structure and hyperparameter settings are included in Appendix E.

**Performances on TU datasets.** As shown in Table 2, SR-GCL achieves the best average accuracy score of 77.03%. Specifically, it achieves 2 bests and 2 second bests on 4 social network datasets: COLLAB, RDT-B, RDT-M5K and IMDB-B, verifying generalization ability of SR-GCL in various types of domain and the effectiveness of GraphBigBird largely reducing the computational complexity while maintaining powerful representation ability. The overall performance of SR-GCL in Table 2 demonstrates its promising performance on both biochemical and social network domains.

## 3.3 EFFECTIVENESS OF SELF-ATTENTIVE RATIONALIZATION

We pick out a molecule from pre-training dataset – ZINC-2M at random for visualization. Figure 6(a) and 6(b) show the original graph and corresponding attention map generated by the pre-trained model, respectively. To be specific, the aforementioned parameterized probability value $p(v_i|g)$ and $p(e_{jk}|g)$, where $v_i$ is the $i$-th node and $e_{jk}$ is the edge between node $j$ and $k$ in 6(a), correspond to entry $(\langle \text{CLS} \rangle, i)$ and $(j, k)$ in 6(b). To make it more intelligible, the attention weights are mapped to the molecule graph in 6(c), on top and bottom of which are node- and edge-wise view (*i.e.,* $p(v|g)$ and $p(e|g)$), respectively. 6(d) gives the calculation of each atom to Gasteiger partial charge (top) and partition coefficient (bottom), which indicate the asymmetric distribution of electrons in chemical bonds and the lipophilicity/hydrophilicity (red/blue shades) of the molecule, respectively. The shade of each atom reflects its impact on chemical properties – instance-discriminative power. The comparison of 6(c) and 6(d) shows high precision of our proposed self-attentive rationalization, justifying its reliability. We also provide more visualization results in Appendix H.

## 4 CONCLUSION

In this paper, we draw inspiration from the self-attention mechanism in transformers and propose a novel framework for transformer-based graph models, Self-attentive Rationale guided Graph Contrastive Learning (SR-GCL), which reshape the separated design of rationale generator and encoder in existing rationale-aware GCL frameworks. In SR-GCL, the self-attention values in transformer module are leveraged to guide the construction of both node- and edge-wise rationale-aware contrastive pairs, making (1) the graph transformations diverse, and (2) the rationale generator naturally integrated with encoder. Visual inspections on real-world molecule datasets demonstrate the effectiveness of self-attentive rationalization proposed by us and extensive experiments on downstream tasks show that SR-GCL empowers the backbone model with the ability of yielding instance-discriminative representations, thus setting the new state-of-the-art for graph pre-training.

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

# A RELATED WORK

## A.1 GRAPH CONTRASTIVE LEARNING

As a prevalent line of self-supervised learning (SSL) (Chen et al., 2020; He et al., 2022), GCL has been a rising topic among researchers in the graph area. It pre-trains the backbone model to yield instance-discriminative representations by contrasting augmented views of graph samples against each other in large label-free datasets, thus benefiting the downstream supervised fine-tuning. The typical paradigm of existing GCL frameworks consists of two modules: (1) graph-structure data augmentation and (2) contrastive optimization.

### A.1.1 GRAPH AUGMENTATION

Graph augmentation transforms a graph in two correlated but different views. Early studies (You et al., 2020; 2021; Qiu et al., 2020; Zhu et al., 2020) instantiate it by random mechanisms, which corrupt the graph structures or attributes with uniform probability. For example, GraphCL (You et al., 2020) constructs augmented views with strategies of node dropping, edge perturbation, or attribute masking in a random fashion. JOAO (You et al., 2021) adaptively selects graph augmentation strategies, but all alternatives are still confined in random mechanisms. GCC (Qiu et al., 2020) augments the $r$-ego network of nodes by random walks with restarts. And GRACE (Zhu et al., 2020) performs random edge removal and node feature masking to generate diverse views. However, overlooking the importance of keeping saliency properties during augmentation may result in potential loss of semantic information, consequently misleading following contrastive optimization and undermining the goal of instance-discrimination.

**Rationale-guided Augmentation.** To tackle this problem, various methods (Zhu et al., 2021; Liu et al., 2022) have been proposed to preserve salient semantics in augmented views, which we term as rationale-guided augmentation. A research line leverages external domain knowledge to bridge the potential semantic gap between augmented views. For instance, in the social network domain, GCA (Zhu et al., 2021) proposes to utilize node centrality measures to capture crucial connective structures; in the biochemical molecule domain, GraphMVP (Liu et al., 2022) conducts contrast between 2D and 3D conformation structure of molecules. Still, expertise could be expensive or even inaccessible in some scenarios (Tang et al., 2014), making their application very limited and the performances of generalizing to unseen domains are easily damaged by these over-specific guidance (Wang et al., 2022). Aforementioned limitations call for rationale-aware but automatic augmentation mechanisms to get rid of the domain knowledge.

**Rationale Generator.** More recently, GCL frameworks (Suresh et al., 2021; Li et al., 2022; Xu et al., 2022) adopt auxiliary models, which is named rationale generator, to conduct heuristic rationale discovery. Specifically, AD-GCL (Suresh et al., 2021) hires another graph model as the rationale generator to identify salient edges for edge-wise views. RGCL (Li et al., 2022) captures semantically informative substructures to create node-wise views with a GNN-based rationale generator. However, the limited perspective of only node or edge confines the performance of the pre-trained model to a sub-optimal position, which is consistent with the observations in (Chen et al., 2020; You et al., 2020) – "no single transformation suffices to learn good representations". To improve the diversity of views, a naive solution is to adopt multiple auxiliary generators, each of which specialises one perspective. Nonetheless, you cannot have your cake and eat it – by this way, more diversity on rationale-aware transformations demands more system complexity (*i.e.,* more auxiliary generators).

**Rationale Encoder.** Apart from the limited perspectives, existing rationale-aware GCL frameworks separate the rationale generator from the encoder (*i.e.,* the backbone model being pre-trained and fine-tuned). The back-propagation of gradients to the encoder is natural when optimizing the contrastive loss of representations yielded by it. Nevertheless, to enable the propagation to the generator, AD-GCL (Suresh et al., 2021) leverages the Gumbel-Max reparameterization, which may lead to instability stemming from a large variance of gradients (Jang et al., 2017). RGCL (Li et al., 2022) mixes the outputs of the encoder and generator to yield the graph representations subsequently being contrasted, resulting in the biased output of the encoder – the generator will be discarded after pre-training. Here, we blame the separation of the generator and encoder for the aforementioned instability and biased graph-level representations.

### A.1.2 CONTRASTIVE OPTIMIZATION

With contrastive pairs in hand, GCL performs instance-discrimination task by optimizing contrastive loss, which can be instantiated as NCE (Wu et al., 2018b), InfoNCE (van den Oord et al., 2018) or NT-Xent (Chen et al., 2020). During this, the agreement between two augmented views of the same anchor graph is encouraged, while the divergence between those of different anchors is enforced.

### A.2 GRAPH TRANSFORMER

Transformer has achieved remarkable success in various domains (Vaswani et al., 2017; Devlin et al., 2019; Dosovitskiy et al., 2021). Researchers also apply it in graph area (Dwivedi & Bresson, 2020; Kreuzer et al., 2021; Mialon et al., 2021; Ying et al., 2021; Wu et al., 2021; Chen et al., 2022; Rampásek et al., 2022), aiming to take advantages of self-attention mechanism to capture global relations and avoid over-smoothing during the graph representation learning. However, graph is not naturally in agreement with the input format of transformers, but a data structure with topology and high regularity. Thus, early studies transform a graph into a sequence of nodes and only implicitly incorporate topological information by adding absolute (Dwivedi & Bresson, 2020; Kreuzer et al., 2021), relative (Mialon et al., 2021) or structural (Ying et al., 2021) encoding.

More recently, hybrid structures have been proposed to integrate GNN and transformer. In Graph-Trans (Wu et al., 2021), the GNN module first learns local structure to generate token features, and then the transformer module focuses on capturing global relationships. SAT (Chen et al., 2022) leverages a GNN to encode structure information, while a transformer captures interactions between nodes. More recently, GPS (Rampásek et al., 2022) provides a comprehensive recipe for hybrid models, including transformer structures and encoding methods.

## B ALGORITHM OF SR-GCL

---

**Algorithm 1** SR-GCL algorithm

---

1: **Initialize:** dataset $\{g_m : m = 1, 2, ..., M\}$, GNN module $f_1(\cdot)$, transformer module $f_2(\cdot)$, projector $h(\cdot)$, sampling ratio $\rho_v$ & $\rho_e$ and temperature $\tau$.

2: **for all** sampled minibatch of data $\{g_n : n = 1, 2, ..., N\}$ **do**

3:      **for** $n = 1$ **to** $N$ **do**

4:          $\mathbf{Z} = f_1(g_n)$          ▷ Equation 1

5:          $\{p(v|g_n) = \dfrac{\kappa(\mathbf{z}_{\mathrm{CLS}}, \mathbf{z}_v)}{\sum_{u \in \mathcal{V}} \kappa(\mathbf{z}_{\mathrm{CLS}}, \mathbf{z}_u)} : v \in \mathcal{V}\}$          ▷ Equation 5

6:          $\{p(e|g_n) = \dfrac{\kappa(\mathbf{z}_{u_e}, \mathbf{z}_{v_e})}{\sum_{e' \in \mathcal{E}} \kappa(\mathbf{z}_{u_{e'}}, \mathbf{z}_{v_{e'}})} : e \in \mathcal{E}\}$          ▷ Equation 5

7:          $R_v(g) \sim P_{\mathcal{V}}(\cdot | G = g_n))$    s.t. $|\mathcal{V}_R| = \rho_v \cdot |\mathcal{V}|$          ▷ Equation 8

8:          $R_e(g) \sim P_{\mathcal{E}}(\cdot | G = g_n)$    s.t. $|\mathcal{E}_R| = \rho_e \cdot |\mathcal{E}|$          ▷ Equation 9

9:          $\mathbf{r}'_n = h(f_2(f_1(R_v(g)))), \quad \mathbf{r}''_n = h(f_2(f_1(R_e(g))))$

10:      **end for**

11:      **for** $n = 1$ **to** $N$ **do**

12:          $\mathcal{R}_n^- = \{\mathbf{r}'_i, \mathbf{r}''_i : i = 1, 2, ..., n-1, n+1, ..., N\}$

13:          $l^n = -\log \dfrac{\exp\left(\mathbf{r}'_n{}^\top \mathbf{r}''_n / \tau\right)}{\exp\left(\mathbf{r}'_n{}^\top \mathbf{r}''_n / \tau\right) + \sum_{\mathbf{r}^- \in \mathcal{R}_g^-} \exp\left(\mathbf{r}'_n{}^\top \mathbf{r}^- / \tau\right)}$          ▷ Equation 12

14:      **end for**

15:      Update $f(\cdot)$, $r(\cdot)$, and $h(\cdot)$ to minimize $\mathcal{L} = \dfrac{1}{N} \sum_{n=1}^{N} l^n$

16: **end for**

17: **return** Encoder $f_2 \circ f_1$

---

# C    DATASETS

Table 3: Statistics for ZINC-2M and MoleculeNet datasets.

| DATASETS | CATEGORY | UTILIZATION | GRAPHS# | AVG. N# | AVG. E# |
|---|---|---|---|---|---|
| ZINC-2M | BIOCHEMICAL MOLECULES | PRE-TRAINING | 2,000,000 | 26.62 | 57.72 |
| BBBP | BIOCHEMICAL MOLECULES | FINETUNING | 2,039 | 24.06 | 51.90 |
| TOX21 | BIOCHEMICAL MOLECULES | FINETUNING | 7,831 | 18.57 | 38.58 |
| TOXCAST | BIOCHEMICAL MOLECULES | FINETUNING | 8,576 | 18.78 | 38.52 |
| SIDER | BIOCHEMICAL MOLECULES | FINETUNING | 1,427 | 33.64 | 70.71 |
| CLINTOX | BIOCHEMICAL MOLECULES | FINETUNING | 1,477 | 26.15 | 55.76 |
| MUV | BIOCHEMICAL MOLECULES | FINETUNING | 93,087 | 24.23 | 52.55 |
| HIV | BIOCHEMICAL MOLECULES | FINETUNING | 41,127 | 25.51 | 54.93 |
| BACE | BIOCHEMICAL MOLECULES | FINETUNING | 1,513 | 34.08 | 73.71 |

Table 4: Statistics for unsupervised learning TU-datasets.

| DATASETS | CATEGORY | GRAPHS# | AVG. N# | AVG. E# |
|---|---|---|---|---|
| NCI1 | BIOCHEMICAL MOLECULES | 4,110 | 29.87 | 32.30 |
| PROTEINS | BIOCHEMICAL MOLECULES | 1,113 | 39.06 | 72.82 |
| DD | BIOCHEMICAL MOLECULES | 1,178 | 284.32 | 715.66 |
| MUTAG | BIOCHEMICAL MOLECULES | 188 | 17.93 | 19.79 |
| COLLAB | SOCIAL NETWORKS | 5,000 | 74.49 | 2457.78 |
| RDT-B | SOCIAL NETWORKS | 2,000 | 429.63 | 497.75 |
| RDT-M | SOCIAL NETWORKS | 4,999 | 508.52 | 594.87 |
| IMDB-B | SOCIAL NETWORKS | 1,000 | 19.77 | 96.53 |

# D    BASELINES

We compare with following baselines to verify the state-of-the-art performance of our SR-GCL:

- **Infomax** (Velickovic et al., 2019) Infomax conducts mutual information maximization between patches (*i.e.,* subgraphs centered around nodes of interest) and high-level graphs.

- **EdgePred** (Hamilton et al., 2017) EdgePred pre-trains a backbone model by generating node representations of previously unseen data with text attributes to make edge predictions.

- **Attribute Masking** (Hu et al., 2020) Attribute Masking masks attributes of node/edge and make backbone to predict them based on their neighboring structure.

- **Context Prediction** (Hu et al., 2020) Context Prediction trains the backbone to map nodes appearing in similar contexts to nearby representations by predicting with subgraphs.

- **GraphCL** (You et al., 2020) GraphCL is a vanilla GCL method, constructing augmented views with strategies of node dropping, edge perturbation or attribute masking in a random fashion.

- **GraphLoG** (Xu et al., 2021) GraphLoG preserves the local similarity and leverages prototypes to capture the clusters with global semantics.

- **AD-GCL** (Suresh et al., 2021) With a GNN-based augmenter, AD-GCL adopts adversarial graph augmentation strategies to minimize redundant information in GCL.

- **JOAO** (You et al., 2021) JOAO automatically selects data augmentations from a strategy pool to construct contrastive pairs for GCL.

- **RGCL** (Li et al., 2022) RGCL hires an auxiliary model to reveal semantically important nodes to construct augmented views for contrastive optimization.

- **graph2vec** (Narayanan et al., 2017) Graph2vec views the graph as a document and the rooted subgraphs around every node as words that compose the document. Then the document embedding neural networks can be adopted to learn representations of graphs

- **Infograph** (Sun et al., 2020) Infograph maximizes the mutual information between the graph-level representation and the representations of substructures.

- **GraphMAE** (Hou et al., 2022) GraphMAE is a masked graph autoencoding framework aiming at masked feature reconstruction.

## E  BACKBONE MODEL ARCHITECTURES

Table 5: Backbone model architectures and hyperparameters

| MODELS | GIN (T) | GRAPH TRANS (T) | GRAPH TRANS (U) |
|---|---|---|---|
| GNN TYPE | GIN | GIN | GIN |
| NEURON# | [300,300,300,300,300] | [300,300,300,300,300] | [32,32,32] |
| TRANS IN/OUT DIM | - | 128 | 32 |
| TRANS PROJ DIM | - | 512 | 64 |
| TRANS LAYER# | - | 4 | 2 |
| TRANS HEADS# | - | 4 | 2 |
| POOLING | MEAN | $\langle$CLS$\rangle$ | $\langle$CLS$\rangle$ |
| PROJECTOR NEURON# | [300,300] | [128,128] | [32,32] |
| OPTIMIZER | ADAM | ADAMW | ADAMW |
| LEARNING RATE | 0.001 | 0.0001 | 0.0001 |

(T) means that the model is applied in transfer learning experiments, while (U) indicates unsupervised learning experiments. For hyperparameter sensitivity analysis in transfer learning, please refer to Figure 7. And we only adopt GraphBigBird on three datasets containing graphs of thousands of nodes: DD, RDT-B and RDT-M and the default value for sparsity $r$ is set to 0.01.

## F  COMPUTATIONAL EFFICIENCY

The training time mainly depends on the type of backbone model. To compare the computational efficiency of commonly adopted GIN model in existing baselines with the graph transformer in our paper, the architectures of which are shown in Table 5, we evaluate the number of parameters and the propagation runtime in transfer learning for each mini-batch of 256 samples on NVIDIA GeForce RTX 3090. The statistics in Table 6 show that the transformer module adds an overhead of 46.63% to whole contrastive learning procedure and 49.91% to inference (*i.e.,* forward) time, which we argue, considering the great improvement of representation ability and interpretability, is acceptable.

Table 6: Computational efficiency of backbone models

| Backbone | Forward (ms) | Backward (ms) | Cumulative (ms) | Parameter # |
|---|---|---|---|---|
| GIN | $15.55_{\pm9.72}$ | $22.49_{\pm2.27}$ | $38.04_{\pm10.07}$ | 1.88M |
| Graph Transformer | $23.97_{\pm11.39}$ | $36.51_{\pm3.32}$ | $60.47_{\pm11.95}$ | 2.71M |

Meanwhile, we conduct the time complexity comparison between SR-GCL and the vanilla GCL – GraphCL with the same graph transformer as the backbone model, which is shown as follows:

Table 7: Time complexity comparison between SR-GCL and GraphCL

| FRAMEWORKS | GRAPHCL | SR-GCL |
|---|---|---|
| DATA AUG. | $O(2B\rho|\mathcal{V}|\log|\mathcal{V}|)$ | $O(B(\rho_v|\mathcal{V}|\log|\mathcal{V}| + \rho_e|\mathcal{E}|\log|\mathcal{E}|))$ |
| RATIONALE GEN. | — | $O(B((|\mathcal{E}|^2 + |\mathcal{V}|)L_{\text{GNN}}) + |\mathcal{V}|^2d + |\mathcal{V}|d^2)$ |
| BACKBONE FORWARD | $O(B((2|\mathcal{E}|^2 + |\mathcal{V}|)L_{\text{GNN}} + (|\mathcal{V}|^2d + |\mathcal{V}|d^2)L_{\text{GT}}))$ | $O(B((2|\mathcal{E}|^2 + |\mathcal{V}|)L_{\text{GNN}} + (|\mathcal{V}|^2d + |\mathcal{V}|d^2)L_{\text{GT}}))$ |
| CONTRASTIVE LOSS | $O(B^2d)$ | $O(B^2d)$ |

where the average numbers of nodes and edges per graph in ZINC-2M datasets are $|\mathcal{V}|$ and $|\mathcal{E}|$, respectively, $B$ denotes the batch size, $\rho$ denotes the sampling ratio, $L$ denotes the number of layers in backbone encoder, and $d$ denote the latent space dimension where contrastive loss is calculated. On a single NVIDIA GeForce RTX 3090, the overall training time of GraphCL and SR-GCL with 100

epochs and the same backbone model of graph transformer are 25.61 and 30.56 hours, respectively. We argue that the 19.33% training time overhead brought by self-attention rationalization process is acceptable, considering the performance gain of SR-GCL compared with GraphCL.

# G HYPERPARAMETER SENSITIVITY

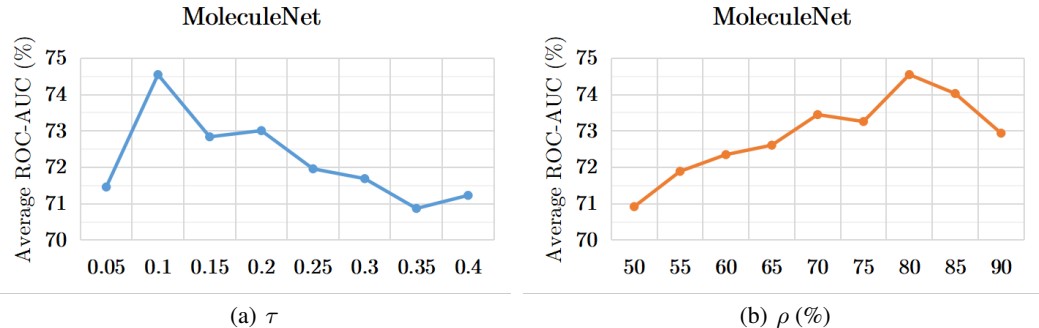

(a) $\tau$            (b) $\rho$ (%)

Figure 7: Sensitivity *w.r.t.* hyperparameters $\tau$ and $\rho$.

The average performances of SR-GCL for transfer learning on 8 downstream MoleculeNet datasets *w.r.t.* hyperparameters $\tau$ and $\rho$ are presented in Figure 7. The sharp pike in 7(a) indicates the close relationship between SR-GCL's performance and hyperparameters $\tau$ (Wang & Liu, 2021). Thus, we suggest to tune it carefully around 0.1. As for hyperparameter controlling the scale of augmented views, we set $\rho_v = \rho_e = \rho$ to make the scales of each contrastive pair close. The results show that SR-GCL achieves the best performance when $\rho$ is set to a large value around 80%, which is consistent with Li et al. (2022).

# H  SAMPLES OF SELF-ATTENTIVE RATIONALIZATION

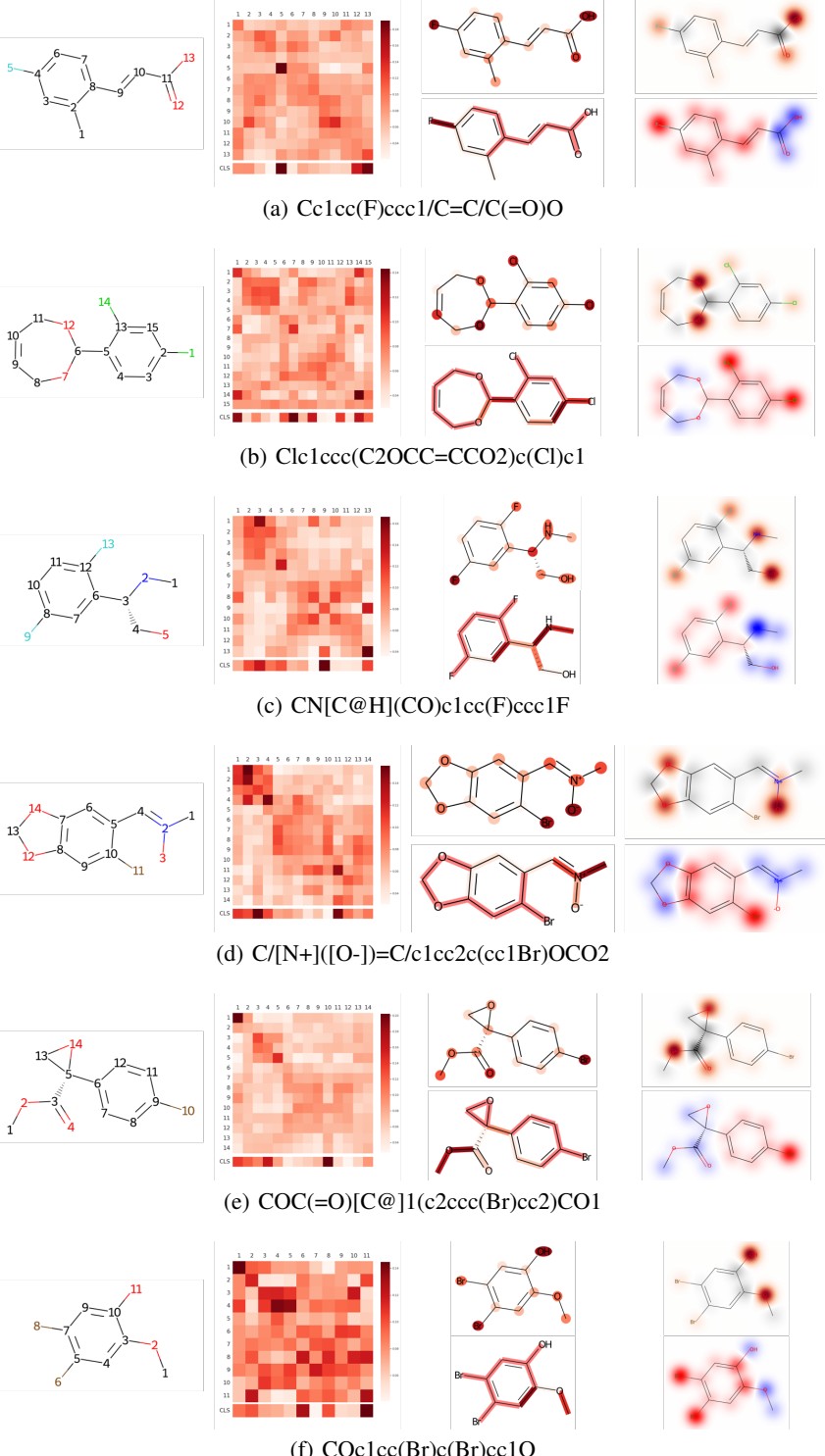

Figure 8: Visualization of more molecules

