# OpenReview forum: "Self-attentive Rationalization for Graph Contrastive Learning"
_ICLR.cc/2023/Conference — Submitted to ICLR 2023_

### Official Review · Reviewer_YpSx · 2022-10-21

**Confidence:** 4
**Correctness:** 3
**Technical Novelty And Significance:** 3
**Empirical Novelty And Significance:** 2
**Recommendation:** 5

**Clarity, Quality, Novelty And Reproducibility:**

- The proposed method is new but still requires more validation whether the whole pipeline requires much higher computational cost during pre-training.
- The clarity can be improved, specifically for the term selection as mentioned above.


**Strength And Weaknesses:**

Strength:
- Depending on how the graph structures are constructed, different edges certainly contain different levels of information on how the neighboring nodes are related for different tasks. Leveraging additional information for helping identify the importance of the relationship intuitively should help build a more meaningful corrupted view of graphs for better graph contrastive learning.
- The proposed modification of conventional graph contrastive learning is straightforward and easy to follow.
- Figure 3 and Figure 4 are helpful in comprehending how the algorithm works.
- Sensitivity w.r.t hyper-parameters analysis in Figure 6 is appreciated.

Weaknesses
- It is not clear how the transformer is obtained for providing the attention weights. If the transformer needs to be pre-trained before the graph contrastive self-supervised pre-training, then the whole process would involve 2 large-scale pre-training stages, one for obtaining the transformer, and another one for the actual graph pre-training. If that is the case, the proposed approach requires significantly more computational cost than the compared baseline methods.
- As mentioned in the limitation section, the current method has been tested on graph-level classification tasks only. However, node classification or edge classification offer more flexibility for practical use cases, for example social network connectivity identification or fraud detection for anti-money laundering. It is important to have signals on those tasks before we can faithfully validate the proposed method works effectively and consistently.
- The writing is good in the manuscript. However I would appreciate the authors using terms that are simpler and easier to understand. For example, the authors use the term “rational finder” but in fact it is simply using the attention weight for getting the importance values. That way, junior readers can have a better understanding of the proposed method quicker.


**Summary Of The Paper:**

The authors propose to leverage the attention weights from a pre-trained transformer for guiding how the nodes and edges are sampled during graph contrastive learning. In contrast to previous work that often performs node or edge dropping for creating different views of a graph, the proposed method claims that the stochastic version by sampling via attention weights provides more meaningful views compared to the uniformly sampled ones. Experiments show improvement on MoleculeNet for graph-level classification tasks.

**Summary Of The Review:**

- The paper can be largely improved by providing node-level and edge-level classification results on more benchmarks. Also would be great to clarify whether the proposed pipeline requires 2 stages of pre-training.

---

> ### Author Response · Authors · 2022-11-18
> **Response to Reviewer YpSx**
>
> Thanks for your valuable comments. Following your suggestions, we have carefully revised our paper to make the pipeline of SR-GCL clearer and term usage consistent with existing baselines. To address your concerns, detailed point-to-point responses are provided below.
>
> **Comment 1: It is not clear how the transformer is obtained for providing the attention weights.**
>
> Response 1: Here we clarify that SR-GCL is a one-stage pre-training scheme. The GNN and transformer module are both randomly initialized at the beginning and simultaneously optimized. The contrastive supervision signal guides the backbone model (both GNN and transformer module) to learn instance-discriminative representations, which indicates that the self-attention mechanism in the transformer is enhanced concurrently, thus bringing about more accurate self-attentive rationalization. And please check the algorithm in Appendix B for the pipeline of SR-GCL. We have carefully reorganized the paper, especially the method chapter, to make it easier to follow.
>
>
>
> **Comment 2: SR-GCL is tested on graph-level classification tasks only.**
>
> Response 2: Thanks for pointing that out. We respectfully argue that SR-GCL indeed focuses on graph-level semantic information, but there is a research gap between node-level and graph-level GCL methods, which can be summarized as:
>
>
>
> |                 | Contrast types                   | Optimization goal          | Downstream tasks     | Potential  applications                                 | Examples                                  |
> | --------------- | -------------------------------- | -------------------------- | -------------------- | ------------------------------------------------------- | ----------------------------------------- |
> | Node-level GCL  | Node-node or node-graph contrast | Node-level representation  | node classification  | anomaly detection, social network analysis              | GCC [6], AFGRL [2]                        |
> | Graph-level GCL | Graph-graph contrast             | Graph-level representation | graph classification | chemical property prediction , assistant drug discovery | SR-GCL, GraphCL [1], AD-GCL [3], RGCL [4] |
>
> The motivation and task formation are quite different and at this point, thus there's no single GCL framework that could perfectly bridge the gap and achieve SOTA on both node- and graph-level tasks. If your time allows, we respectfully invite you to refer to the briefs, motivations and task formation part in Appendix A and the baselines [1,3,4,5], to know this graph-level GCL sub-area.
>
>
>
> **Comment 3: Term selection.**
>
> Response 3: This valuable suggestion is appreciated. We have polished the paper and made the term usage consistent with former baselines (See Method chapter). For example, we replaced "rationale finder" with "rationale generator", which is first used in RGCL [4] to name the module discovering the rationales in graphs.
>
>
>
> [1] You, Yuning, et al. "Graph contrastive learning with augmentations." Advances in Neural Information Processing Systems 33 (2020): 5812-5823.
>
> [2] Lee, Namkyeong, Junseok Lee, and Chanyoung Park. "Augmentation-free self-supervised learning on graphs." Proceedings of the AAAI Conference on Artificial Intelligence. Vol. 36. No. 7. 2022.
>
> [3] Suresh, Susheel, et al. "Adversarial graph augmentation to improve graph contrastive learning." Advances in Neural Information Processing Systems 34 (2021): 15920-15933.
>
> [4] Li, Sihang, et al. "Let invariant rationale discovery inspire graph contrastive learning." International Conference on Machine Learning. PMLR, 2022.
>
> [5] Hu, Weihua, et al. "Strategies for Pre-training Graph Neural Networks." International Conference on Learning Representations. 2019.
>
> [6] Qiu, Jiezhong, et al. "Gcc: Graph contrastive coding for graph neural network pre-training." *Proceedings of the 26th ACM SIGKDD International Conference on Knowledge Discovery & Data Mining*. 2020.

---

> ### Author Response · Authors · 2022-11-29
> **Gentle reminder to Reviewer YpSx**
>
> Dear Reviewer YpSx,
>
> Thank you for your valuable comments and time in reviewing our paper. It would be appreciated if you can confirm whether our responses have addressed your concerns. Always glad to have further discussions on your concerns about our paper.

---

### Official Review · Reviewer_KsiL · 2022-10-23

**Confidence:** 4
**Clarity, Quality, Novelty And Reproducibility:** Clarity, quality, novelty and reprodu…
**Correctness:** 4
**Technical Novelty And Significance:** 4
**Empirical Novelty And Significance:** 3
**Recommendation:** 6

**Strength And Weaknesses:**

Strength:
1. The writing is good.  It is easy to understand the main idea.
2.  The experimental analysis is sufficient and in-depth.

Weakness:
1. More theoretical analysis about the effectiveness of  SR-GCL should be clarified.


**Summary Of The Paper:**

This paper presents a simple and effective  Self-attentive Rationale guided Graph Contrastive Learning  for graph network learning. The code idea is to learn both node- and edge-wise rationale-aware views. Extensive experimental results show the effectiveness of the proposed method.

**Summary Of The Review:**

The proposed solution for graph contrastive learning is simple and clear. The experimental results are sufficient.

---

> ### Author Response · Authors · 2022-11-18
> **Response to Reviewer KsiL**
>
> Thanks so much for your time and positive feedback.
>
> **Comment 1: More theoretical analysis should be clarified.**
>
> Response 1: Good points. From earlier work like SimCLR, MOCO, and BYOL in CV area to GraphCL, JOAO and RGCL in graph area, CL frameworks are mostly empirically demonstrated effective and little theoretical analysis is provided. The research community is calling for more theoretical insights, especially for rationale-aware CL frameworks. We will keep this in mind and see what we can do in the future.

---

> > ### Comment · Reviewer_KsiL · 2022-11-27
> > **About R1**
> >
> > I have no further questions about the current version. I would refer to other reviewers' comments and the author's response before making a decision.

---

> > > ### Author Response · Authors · 2022-11-27
> > > **Response to the feedback**
> > >
> > > Thanks very much for your response. We are looking forward to discussing with you in other reviews' sessions on their concerns.

---

### Official Review · Reviewer_K54f · 2022-10-24

**Confidence:** 4
**Correctness:** 3
**Technical Novelty And Significance:** 3
**Empirical Novelty And Significance:** 3
**Recommendation:** 3

**Clarity, Quality, Novelty And Reproducibility:**

While this work proposes to use attention weights for graph augmentation, the evaluation of combing view generator and the embedding encoder is missing. Furthermore, only one type of dataset is adopted for evaluation. Normally graph contrastive learning methods are evaluated over different types of datasets for more comprehensive experiments.

**Strength And Weaknesses:**

Strengths:
The proposed model utilizes the attention map in self-attention for automatic data augmentation, which improves the model efficiency and potentially improves the model training to be more stable and less prone to overfitting.
Diverse types of views are generated using a shared view generator in the proposed model, which not only improves the efficiency but also enhance the data augmentation in GCL.
The proposed model is evaluated from different dimensions, including illustrative case study on the explainability of the self-attention module.

Weaknesses:
The proposed model does not achieve best performance compared to baselines in some circumstances in Table 1.
The experiments are only conducted on biochemistry datasets, which cannot show the generality of the proposed model. More datasets such as academic citation networks, social networks and recommendation graphs could be utilized.
The authors claim that combining the view generator and the encoder is beneficial for model efficiency, but no experiments targets this point.


**Summary Of The Paper:**

This paper proposes to mitigate two limitations in the existing graph contrastive learning (GCL) frameworks: i) The lack of view diversity in data augmentation approaches degenerates the effectiveness of self-discriminative contrastive learning. ii) Existing methods utilize separated automated view generator and encoder, which damage the efficiency and efficacy of model training for GCL. To address the limitations, the proposed model employs a self-attention module to generate attention map for view generation and encoding simultaneously, to avoid introducing additional view generator.

**Summary Of The Review:**

Considering many recent efforts on graph contrastive learning, this work proposes to leverage self-attention to infer node correlations for graph augmentation. However, the insufficient experiments may not be able to demonstrate the effectiveness of the new introduced approach.

---

> ### Author Response · Authors · 2022-11-18
> **Response to Reviewer K54f**
>
> Thanks for your comments. Below we provide the point-to-point responses to clarify some misunderstandings of our proposed method and we have carefully revised our paper with more experiments on social network datasets  and a clearer analysis of experiments to justify the advantage of our contributions. Looking forward to discussing more with you.
>
> **Comment 1: Not the best on every downstream dataset.**
>
> Response 1: We respectfully argue that SR-GCL achieves 5 bests and 3 second bests (noted that on BACE, the best and second best both belong to GraphLoG but with different backbones, thus SR-GCL can be seen as the second best pre-training scheme) out of 8 downstream datasets, and outperforms existing baselines by a clear margin on the average score (See Table 1), which demonstrate the effectiveness of SR-GCL. In fact, not a single pre-training baseline in Table 1 achieves universal best on all 8 datasets.
>
>
>
> **Comment 2: Lack of dataset diversity**.
>
> Response 2: Thank you for this valuable suggestion. We have added experiments on more types of datasets to show the generality of SR-GCL. Please see the general response and refer to our revised paper with sparse attention mechanism (See Section 2.4) and more results on TU datasets covering both biochemical and social network domains (See Section 3.2).
>
>
>
> **Comment 3: The advantage of combining the rationale generator and the encoder needs to be empirically verified.**
>
> Response 3: Good points. Here we clarify that this can be verified by comparing the performance of SR-GCL with only node-wise views and that of RGCL with the same backbone model.  We take out these two columns (See Table 1):
>
> |                                  | BBBP  | Tox21 | ToxCast | SIDER | ClinTox | MUV   | HIV   | BACE  | Avg.  |
> | -------------------------------- | ----- | ----- | ------- | ----- | ------- | ----- | ----- | ----- | ----- |
> | RGCL with graph transformer      | 71.36 | 75.72 | 63.93   | 60.91 | 80.02   | 75.95 | 77.81 | 79.92 | 73.20 |
> | SR-GCL only with node-wise views | 73.01 | 75.12 | 65.69   | 61.21 | 80.47   | 77.92 | 77.89 | 81.33 | 74.08 |
>
> They have the same graph transformer as the backbone and limited view - node-wise. The difference lies in the rationale generator. RGCL utilizes an external GNN to discover the rationale, while SR-GCL leverages this graph transformer as the combined rationale generator and encoder. The performance gain verifies the claim that combining the view generator and the encoder is beneficial for model efficiency. We have added a more detailed experiments analysis in the revised paper to better clarify our claims (See Section 3.1 Observation 3).

---

> ### Author Response · Authors · 2022-11-29
> **Gentle reminder to Reviewer K54f**
>
> Dear Reviewer K54f,
>
> Thank you for your valuable comments and time in reviewing our paper. It would be appreciated if you can confirm whether our responses have addressed your concerns. Always glad to have further discussions on your concerns about our paper.

---

### Official Review · Reviewer_oC1b · 2022-10-27

**Confidence:** 3
**Correctness:** 3
**Technical Novelty And Significance:** 3
**Empirical Novelty And Significance:** 3
**Recommendation:** 6

**Clarity, Quality, Novelty And Reproducibility:**

Clarity: the presentation of the method is clear, but the running efficiency is unclear.

Novelty: to my knowledge, using self-attention to learn node/edge importance scores for adaptive GCL augmentation is novel.

**Strength And Weaknesses:**

S1. The paper is easy to follow and the idea is clear.

S2. The paper uses case studies to visualize the attention scores to verify the rationality of the learned scores.

W1. The experiments are only conducted on molecule graphs. How about other types of graphs that are widely used in GCL papers (e.g., social networks, Reddit, etc.)? I have personally tested the code for social network datasets without pre-training, and the results show no obvious improvement from baselines such as RGCL. If my test is true, then the practicality of the method is limited, as it needs a large dataset for pre-training, which may not be easy for the graphs except molecule graphs.

W2. Runtime efficiency is not analyzed and experimentally tested

**Summary Of The Paper:**

The paper proposes to use a self-attention mechanism to learn node-level and edge-level importance scores for a graph, and then conduct adaptive sampling considering the node/edge-level scores for GCL. The results have been conducted on 8 molecule datasets.

**Summary Of The Review:**

While the method looks good and sensible, whether it can be generalized for other types of graphs (besides molecule) and the running efficiency is not clarified.

---

> ### Author Response · Authors · 2022-11-18
> **Response to Reviewer oC1b**
>
> Thanks for your valuable and constructive comments. Below we provide the point-to-point response to address your concerns. And we have carefully revised our paper with more experiments on social network datasets and analysis of runtime efficiency. Looking forward to discussing more with you.
>
> **Comment 1: Diversity of datasets.**
>
> Response 1: This comment is highly valued. Please see the general response and refer to our revised paper with sparse attention mechanism (See Section 2.4) and more results on TU datasets (See Section 3.2) covering both biochemical and social network domains. Concretely, SR-GCL achieves the best average accuracy score on 8 TU datasets. Specifically, it achieves 2 bests and 2 second bests on 4 social network datasets, verifying its general effectiveness on datasets of various domains.
>
> And I'm not sure about how you tested the code without pre-training, since SR-GCL is an SSL framework for pre-training and cannot be directly applied on large-scale social network graphs without space optimization due to the quadratic space complexity of the transformer.
>
> **Comment 2: Runtime efficiency is not analyzed and experimentally tested.**
>
> Response 2: Thanks for your valuable suggestion. In our original paper, there is some running time efficiency analysis for backbone models in the appendix. And in this revised version, we additionally added more complexity analysis and runtime evaluation (See Appendix F).

---

> > ### Comment · Reviewer_oC1b · 2022-11-21
> > **Appreciate the feedback**
> >
> > "Sec 3.2 unsupervised learning” is exactly what I mean by 'no pre-training with a large dataset'. I appreciate the authors' feedback and new results. I thus raise my score from 5 to 6.

---

> > > ### Author Response · Authors · 2022-11-27
> > > **Response to the feedback**
> > >
> > > We again appreciate your time in our revised paper and we are always open for any further discussions with you.

---

### Official Review · Reviewer_PbQf · 2022-10-29

**Confidence:** 3
**Correctness:** 3
**Technical Novelty And Significance:** 3
**Empirical Novelty And Significance:** 3
**Recommendation:** 5

**Clarity, Quality, Novelty And Reproducibility:**

In my opinion, I cannot follow this paper well. I'm not sure if it's because it's specific tasks on graph learning. Although I am an expert in graph learning, I cannot understand this paper well in motivation and some claims. The originality is ok.

**Strength And Weaknesses:**

Strength
1. A rationale finder finder is proposed with plural views instead of either node- or edge-wise rationales finder independently.
2. In this paper, based on the transformer framework, they propose self-attentive for dual views to guide the rationale finder.

Weakness:
1. I do not think of “no single transformation suffices to learn good representations‘’. There are many augmentation-free (in raw attributes and the graph) methods working competitively in different benchmarks. I'm curious if augmentation-free is also applicable in specific data.
2 . In my opinion, the contribution is somewhat limited. It looks like the biggest contribution is a self-attentive augmentation. From a GCL point of view, the contribution seems insufficient. Can this technique be used in the other graphs and tasks? For example node classification and social network.
3. I'm not sure I understand correctly that sampling makes this model not an end2end optimization. Have you consider using the reparameterization trick in VAE to solve this? or it is unreasonable or useless.

**Summary Of The Paper:**

In this paper, the authors propose a self-attentive rationalization for rationale-aware agmented pairs, which are used in contrastive learning. The efficiency of self-attentive rationalisation is validated by visualisation on real-world biochemistry datasets, and the performance on downstream tasks shows the good performance of SR-GCL for graph model pre-training.

**Summary Of The Review:**

Based on my comments, I am inclined to give a rejection right now. But I am happy to refer to other reviewers' comments and the author's response before making a decision.

---

> ### Author Response · Authors · 2022-11-18
> **Response to Reviewer PbQf**
>
> Your comments are appreciated. Below we provide the point-to-point response to clarify some misunderstandings of our proposed method and we have carefully revised our paper to make the verification of our claim clearer and the pipeline of our framework easier to follow. Looking forward to discussing more with you.
>
> **Comment 1: ''No single transformation suffices to learn good representation''.**
>
> Response 1: It's quoted from Section 3.2 of the paper SimCLR [1]. Concretely, Figure 5 in SimCLR [1] and Figure 2 in GraphCL [2] share the same observation: contrasting views from different transformations benefits the model -- in the figures, the best performance is achieved at non-diagonal entries instead of diagonal ones. And this is consistent with the results in our ablation study (See Section 3.1 Observation 4) -- dual-view outperforms single-view (node- or edge-wise). There are indeed some augmentation-free GCL methods like AFGRL [3]. It treats similar nodes in a graph as the positive pair to optimize the node-level representations, while SR-GCL focuses on graph-level semantic information. The major differences of them are summarized in the table below:
>
> |                 | Contrast types                   | Optimization goal          | Downstream tasks     | Potential  applications                                 | Examples                                  |
> | --------------- | -------------------------------- | -------------------------- | -------------------- | ------------------------------------------------------- | ----------------------------------------- |
> | Node-level GCL  | Node-node or node-graph contrast | Node-level representation  | node classification  | anomaly detection, social network analysis              | GCC [7], AFGRL [3]                        |
> | Graph-level GCL | Graph-graph contrast             | Graph-level representation | graph classification | chemical property prediction , assistant drug discovery | SR-GCL, GraphCL [2], AD-GCL [4], RGCL [5] |
>
>
> Thus, we respectfully argue that the motivation and task formation are quite different. And at this point, there's no GCL framework that could perfectly bridge the gap and achieve SOTA on both node- and graph-level tasks.
>
> **Comment 2: Is it an end2end optimization?**
>
> Response 2: Yes. At every mini-batch iteration, the self-attention scores are updated first, and then based on which, augmentation views are generated. Finally, the common contrastive optimization is conducted. The contrastive supervision signal guides the backbone model (both GNN and transformer module) to learn instance-discriminative representations, which indicates that the self-attention mechanism in the transformer is enhanced concurrently, thus bringing about more accurate self-attentive rationalization. Please refer to the algorithm in Appendix B to check the pipeline.
>
> And actually, the reparameterization trick is adopted in AD-GCL [4], which unfortunately leads to unstable training and sub-optimal performance -- you can check this by running their official codes on transfer learning experiment. This is also one of our motivations to combine rationale generator and encoder to directly utilize supervision signals.
>
> **Comment 3: Unable to follow.**
>
> Response 3: Thanks for your suggestions. And we have polished the paper to make it easier to follow. Please see highlighted part in the updated version. And if your time allows, we respectfully invite you to refer to the briefs, motivations and task formation part in Appendix A and the baselines [2,4,5,6], to know this graph-level GCL sub-area.

---

> > ### Author Response · Authors · 2022-11-18
> > **Citations in Response**
> >
> > [1] Chen, Ting, et al. "A simple framework for contrastive learning of visual representations." International conference on machine learning. PMLR, 2020.
> >
> > [2] You, Yuning, et al. "Graph contrastive learning with augmentations." Advances in Neural Information Processing Systems 33 (2020): 5812-5823.
> >
> > [3] Lee, Namkyeong, Junseok Lee, and Chanyoung Park. "Augmentation-free self-supervised learning on graphs." Proceedings of the AAAI Conference on Artificial Intelligence. Vol. 36. No. 7. 2022.
> >
> > [4] Suresh, Susheel, et al. "Adversarial graph augmentation to improve graph contrastive learning." Advances in Neural Information Processing Systems 34 (2021): 15920-15933.
> >
> > [5] Li, Sihang, et al. "Let invariant rationale discovery inspire graph contrastive learning." International Conference on Machine Learning. PMLR, 2022.
> >
> > [6] Hu, Weihua, et al. "Strategies for Pre-training Graph Neural Networks." International Conference on Learning Representations. 2019.
> >
> > [7] Qiu, Jiezhong, et al. "Gcc: Graph contrastive coding for graph neural network pre-training." *Proceedings of the 26th ACM SIGKDD International Conference on Knowledge Discovery & Data Mining*. 2020.

---

> ### Author Response · Authors · 2022-11-29
> **Gentle reminder to the reviewer**
>
> Dear Reviewer PbQf,
>
> Thank you for your valuable comments and time in reviewing our paper. It would be appreciated if you can confirm whether our responses have addressed your concerns. Always glad to have further discussions on your concerns about our paper.

---

### Author Response · Authors · 2022-11-18
**General response to all the reviewers**

We gratefully thank all the reviewers for your valuable comments and are encouraged that you find our idea -- self-attention for adaptive GCL augmentation novel (Reviewer oC1b, KsiL, YpSx) and our representation clear and easy to follow (Reviewer oC1b, KsiL, YpSx).

Meanwhile, thank the reviewers (PbQf, oC1b, K54f) for pointing out a major weakness: the limited diversity of datasets. Compared with molecule graphs containing only dozens of nodes, some social network graphs hold thousands of nodes, resulting in OOM problem for vanilla transformer with quadratic space complexity. To mitigate this, we additionally propose GraphBigBird  (See Section 2.4), a tailor-made sparse attention mechanism for graph transformer to accommodate large-scale graphs, and have added experiments (See Section 3.2) to further evaluate SR-GCL on TU datasets covering both biochemical and social network datasets.

We have carefully revised our paper by taking into account all the suggestions and highlighted the revised part in the updated file. And please see the updated file for details and find the point-to-point responses below.

---

### Decision · Program_Chairs · 2023-01-20

**Decision:**

Reject

**Justification For Why Not Higher Score:**

Reviewers have had a number of concerns including incremental results, limited dataset types used in evaluations and problems, limited theoretical analysis and lack of clarity in writing. For these reasons, reviewers have not raised the scores to the sufficient level. Overall, the paper requires another round of reviews and cannot be accepted at this time.

**Justification For Why Not Lower Score:**

N/A

**Metareview: Summary, Strengths And Weaknesses:**

This submission has been evaluated by five reviewers and received 1x3, 2x5 and 2x6 scores. Reviewers have had a number of concerns including incremental results, limited dataset types used in evaluations and problems, limited theoretical analysis and lack of clarity in writing. Overall, the paper requires another round of reviews and cannot be accepted at this time.